# Putting the "Machine" Back in Machine Learning for Engineering Students

**Ting-Wu Chin** [1]  **Dimitrios Stamoulis** [1 2]  **Diana Marculescu** [3]

## Abstract

Computer hardware architecture has played an important role in the recent advances made in deep learning and associated applications. However, effective teaching strategies for hardware architectures for machine learning require a different structure and technical background than classic machine learning. More specifically, not only does the material need to convey necessary machine learning concepts to students, but also cover the hardware and software infrastructure concepts required for supporting machine learning systems. In this paper, we describe our approach to designing the course materials along with student assessment and evaluation for the "Hardware Architectures for Machine Learning" course targeting Electrical and Computer Engineering graduate students.

## 1. Introduction

With the recent advances in deep learning, machine learning has gained tremendous attention from students and young academic trainees. According to the 2021 AI Index Report from Stanford HAI (Zhang et al., 2021), "The number of courses that teach students the skills necessary to build or deploy a practical AI model on the undergraduate and graduate levels has increased by 102.9% and 41.7%, respectively, in the last four academic years." While courses for machine learning are ubiquitous in academic institutions, the hardware aspect of machine learning is receiving much less attention despite its crucial role in advancing deep learning.

Understanding the hardware aspect of machine learning is critical for making machine learning models work for various hardware platforms, which can in turn democratize machine learning to a wider audience. As an example, there are around 250 billion devices based on microcontrollers (mic) while there are merely 80 million personal

computers in the world (pc). On the other hand, building suitable hardware for machine learning can be critical for the energy efficiency of the machine learning systems used for large scale training (Patterson et al., 2021). As a result, it is critical to have courses teaching the hardware aspect of machine learning.

We would be remiss if we did not mention that perhaps the main reason why teaching hardware architectures for machine learning deployment is crucial lies in the deep learning revolution that has taken over the world in last decade: indeed, none of it would have been possible without the advent of hardware platforms exhibiting large scale parallelism that have enabled the exponential growth in development and deployment of machine learning systems.

In this paper, we present our experience in designing the graduate-level course materials along with student assessment and evaluation for the "Hardware Architectures for Machine Learning" class for students majoring Electrical and Computer Engineering. Compared to a introductory machine learning course, we focus on teaching students to analyze the hardware-related metrics for machine learning algorithms with a clear focus on deep learning. To achieve this goal, we have had to introduce not only the machine learning but also the hardware concepts, which has inevitably required us to be thoughtful in what material from a standard machine learning course should be included, and what can become optional. In the following sections, we describe the topic selection process, homework and project design strategy, student feedback from the course, and finally the conclusion.

## 2. Class structure and topics coverage

The "Hardware Architectures for Machine Learning" class was designed as graduate-level class, intended for first-year graduate students or advanced senior undergraduate students. The idea of offering the class came during summer of 2018 after the instructor had already run a pilot of a few lectures and homework assignments using machine learning as an application in a separate graduate-level class on energy efficient hardware design. The feedback from students was very positive, and with the help of several enthusiastic teaching assistants (all of whom were Ph.D. students in their second through fourth year of doctoral studies) the class

---

[*]Equal contribution  [1]Department of ECE, CMU [2]Microsoft Cloud and AI [3]Department of ECE, UT Austin. Correspondence to: Diana Marculescu <dianam@utexas.edu>.

*Proceedings of the $2^{nd}$ Teaching in Machine Learning Workshop*, PMLR, 2021.

came to fruition in Fall 2018 as a first iteration, and offered again in its final form in Spring 2019.

The "Hardware Architectures for Machine Learning" class requires as pre-requisites "Introduction to Machine Learning" (either undergraduate or graduate level) and one of "Hardware Arithmetic for Machine Learning," an undergraduate class focusing largely on computer arithmetic used in data intensive hardware architectures, or "Introduction to Computer Architecture," an upper level undergraduate class that most students interested in computer hardware take. The reason for allowing two different computer hardware pathways to enter the class is to encourage participation from students who may have an interest in circuits or logic design *vs.* computer architecture or system design. Some of the Electrical and Computer Engineering students interested in enrolling had not already taken the "Introduction to Machine Learning" class, so depending on their level of interest or background they were allowed to take it simultaneously while being enrolled in our class. This worked out well, and based on their input, it allowed for better understanding of topics covered in both classes.

This course provides an overview of current advances in hardware architectures that can enable fast and energy efficient machine learning applications from the edge to the cloud. Topics include hardware accelerators, hardware-software co-design, and general or application specific system design and resource management for machine learning applications. The course requires no textbook and relies only on technical papers available publicly. The grading algorithm included three components to assess student learning and evaluate progress. Specifically, it is separated into *homework* (30%), *paper presentation and discussion* (15%), and *project* (55%). To encourage discussion in paper presentation, we allocate 5% out of the 15% to be discussion. As for project, we have 15% allocated to each of the first two reports and another 25% of the final report.

The primary purpose of the homework assignments was to help students master the material and prepare for the projects. We encouraged students to work together with their classmates to help them understand the basic concepts. However, students were required to do their own homework, unless teaming was explicitly allowed in certain assignments. Homework assignments were due in the evening of the due date. No late homework assignments were accepted.

The project was designed to: (i) help students understand and synthesize all of the course concepts; (ii) demonstrate their ability at correctly stating and implementing the project's goals; (iii) demonstrate their ability to explore and incorporate good engineering trade-offs in a system/subsystem implementation. All project components should have clearly identified the individual contributions of each team member. Any project proposal, report or presentation could have been submitted up to 5 days late, but subject to a 10% per day late penalty.

The topics covered in the course were chosen to cover all aspects of basic supervised and unsupervised machine learning and their hardware implementation implications. The lectures were designed to cover material discussed and presented in recently published technical papers in the area, while assignments, paper presentations, and projects were designed to reinforce concepts and enable hands-on learning. A typical schedule of lecture topics, homework assignments, paper presentations, and project reports is shown in Table 1. In this course, we mainly focus on supervised convolutional neural networks with one lecture on classic supervised learning approaches such as linear and logistic regression and support vector machine, and one lecture on unsupervised learning that focuses on the K-means algorithm. For each of the machine learning topic, we discuss the corresponding hardware architectures in the literature.

## 3. Homework design

We have designed a total of six homework for the course with each homework being related to the material of the ongoing lectures. We split the six homework assignments into three paper reading and three implementation assignments. The inclusion of paper reading has enabled coverage of broad topics in hardware architectures for machine learning and hardware-aware machine learning, both of which are active fields of research. By including paper reading assignments, students were exposed to state-of-the-art methods and were able to absorb new knowledge from papers.

### 3.1. Design of Implementation-based Homework Assignments

The goal of implementation-based homework assignments is to strengthen students' capabilities for implementing modern machine learning models, as well as help students learn the tools to explore the hardware support for machine learning models. We gradually guided the students to learn to implement Convolutional Neural Networks (CNNs) in PyTorch, use hardware architecture models, and finally optimize both the hardware and the model to achieve the best overall performance. To facilitate students' understanding, one of the key learning strategies we found useful for our students was visualizing the empirical data obtained from each of the assignments. Visualization can aid students' understanding by having them reason and explain why certain plots look the way they do and what general conclusions can be drawn from those behaviors.

**CNN Implementation** The first homework assignment involved the implementation of the well-known LeNet network (LeCun et al., 1998) with the MNIST dataset (LeCun

*Table 1.* Course schedule. We start with supervised learning, dive into deep learning, and close with unsupervised learning. HW: Hardware; DNN: Deep Neural Networks; FPGA: Field-programmable gate array.

| LECTURE | HOMEWORK | PROJECT | PAPER PRES. |
|---|---|---|---|
| INTRO TO HW ARCHITECTURES FOR ML | | | |
| HW ARCHITECTURES FOR SUPERVISED LEARNING | | | |
| TOOLS FOR DEEP LEARNING (DL) | 1 OUT | PROJECT TOPICS OUT | |
| DNN OVERVIEW AND IMPLICATIONS IN HW | | | |
| PAPER PRESENTATIONS | 1 DUE/2 OUT | | SESSION I |
| DNN LATENCY: WHERE DO THE CYCLES GO? | | PROJECT SELECTION DUE | |
| DNN ENERGY EFFICIENCY: WHERE DO THE JOULES GO? | | | |
| PAPER PRESENTATIONS | | | SESSION II |
| CUSTOM HARDWARE ARCHITECTURES FOR DNNs | 2 DUE/3 OUT | | |
| DNNs COMPRESSION FOR EFFICIENT HW IMPLEMENTATION | | | |
| PROJECT PRESENTATIONS I | | GROUP 1 - 1ST REPORT DUE | |
| PROJECT PRESENTATIONS II | | GROUP 2 - 1ST REPORT DUE | |
| PAPER PRESENTATIONS | | | SESSION III |
| LOW & VARIABLE PRECISION ARCHITECTURES FOR DNNs | 3 DUE/4 OUT | | |
| FPGA-BASED ARCHITECTURES FOR DNNs | | | |
| PAPER PRESENTATIONS | | | SESSION IV |
| HARDWARE ARCHITECTURE-DNN MODEL CO-DESIGN | | | |
| STORAGE EFFICIENT ARCHITECTURES FOR DNNs | 4 DUE/5 OUT | | |
| PROJECT PRESENTATION I | | GROUP 2 - 2ND REPORT DUE | |
| PROJECT PRESENTATION II | | GROUP 1 - 2ND REPORT DUE | |
| DNNs ON MOBILE ARCHITECTURES | | | |
| PAPER PRESENTATIONS | | | SESSION V |
| EDGE-SERVER SOLUTIONS FOR DNNs | 5 DUE/6 OUT | | |
| HW ARCHS FOR DISTRIBUTED, FEDERATED LEARNING | | | |
| HW ARCHS FOR UNSUPERVISED LEARNING | | | |
| PAPER PRESENTATIONS | | | SESSION VI |
| FINAL PROJECT POSTERS AND DEMOS | 6 DUE | FINAL REPORT DUE | |

& Cortes, 2010) using PyTorch (Paszke et al., 2019). As a starting point for students, we provided a boilerplate code for training a standard LeNet on MNIST using PyTorch. In the homework questions included to assess learning, we asked students to try various hyperparameters involved in training as a hands-on experience for understanding the sources of randomness in modern machine learning systems. Furthermore, we have asked students to identify the number of floating point operations (FLOP) needed to carry out an inference, which was the first step in building up students' awareness that the computation intensity of the machine learning model is as important as its final predictive accuracy. To facilitate students' understanding, we have asked the students to visualize the experimental data, including accuracy *vs.* FLOP, FLOP *vs.* runtime, and accuracy *vs.* runtime for models characterized by different hyperparameters.

**Hardware Modeling** In our second implementation-based homework, we guide the students to understand a model built upon a CNN hardware accelerator (Gao et al., 2017). We have provided a Python environment for the students which includes boilerplate code to interact with the hardware models. More specifically, we build our boilerplate upon the official Github repository[1] for Tetris (Gao et al., 2017). In the assignment, we ask students to first understand the hardware architecture by reading the reference papers and guide them with reasoning questions about the content. We also ask students to change the boilerplate code to reflect different hardware architecture designs and their resulting performance. Similar to the previous assignment, students were asked to visualize the data to help them

further interpret empirically the significance of the results. Specifically, one of the items required was visualizing the trade-offs between throughput and the resulting design area of a possible hardware accelerator implementing the model, given certain design knobs.

**CNN and Hardware Co-exploration** The last implementation-based assignment offers a synergy between the first two assignments where we guide the students to alter neural architectures to observe the resulting impact on a fixed hardware and also alter the hardware architecture given a predefined CNN. In addition, we ask students to explore changing both hardware and neural architectures by visualizing the resulting performance metrics. Specifically, we ask students to provide scatter plots for the solutions comparing model's predictive performance and execution time. Finally, students perform random search-based optimization to identify a good CNN-Hardware implementation pair and reason about the effectiveness of the obtained solution.

### 3.2. Design of Reading-based Homework Assignments

The reading-based assignments are designed to improve students' capability in understanding fundamentals and absorbing knowledge from recent technical papers. Similar to the implementation-based assignments, the chosen papers align with the lecture material. To achieve this, for each subject, we identify relevant papers and split them into two categories. The first category includes topics to be covered in the class lectures, while the second category is used for the assignments. To aid students in absorbing the main technical content conveyed in the paper, our assignments consist

---

[1]https://github.com/stanford-mast/nn_dataflow

of various questions that encourage students to follow along.

In addition, we have also included a few (*e.g.*, one to two) questions to aid students think critically about the papers they read. Usually these questions start with a scenario of a potential failure mode for the proposed method described in the paper and guide the students to generalize the failure mode in the described scenarios. As an example, we ask students to read PuDianNao (Liu et al., 2015) and one of the questions for the students is shown below.

> Section 6.4 claims that "the efficiency and accuracy of scalable-effort classifiers is a strong function of $\delta$, which can be easily adjusted at runtime to an appropriate value." Consider the scenario where the input data characteristics change over time, then the optimal $\delta$ should probably also change. Can you propose an algorithm to address this problem?

One thing to note is that the papers included in the reading assignments included a balanced mix of hardware and machine learning content so as to not overwhelm students whose background was lacking in one aspect or another. To achieve this, the assignments point to specific sections in the paper that students should read carefully, which makes for a more tenable learning experience, even if the students cannot grasp every little detail described in the paper.

## 4. Paper presentations

To aid students in becoming familiar with the state-of-the-art in this area, we have included paper presentation and discussion sessions. In each session, four to five students present the paper of their choice and the rest of the class is required to participate by asking questions or participate in the discussion. We provided students with a list of papers for them to choose from and they selected the papers they want to present in the beginning of the semester. Per our past experience, students are reluctant to sign up for a presentation early in the semester, so to incentivize early participation we offer a few points of extra credit for such sign ups. As shown in Table 1, paper presentation sessions are scattered uniformly throughout the course and support in topic coverage the material presented in lectures.

## 5. Project design

The goal of projects is to take students through a hardware-aware machine learning project experience, which exposes them to the life cycle of a complete hardware architecture design for machine learning. This includes motivation, problem definition, solution, and presentation. We provide four checkpoints for a project, which includes proposal, first report, class presentation, and final report. Since students have various backgrounds coming in the course, we have

provided a wide variety of predefined project topics for students to select from, which has greatly helped students narrow down a project based on their background, experience, and interest. We provide an example below.

> *Topic*: Implementation of the idea of "Adaptive Neural Network for Efficient Inference (Bolukbasi et al., 2017)" with current state-of-the-art deep neural networks such as NasNet (Zoph et al., 2018) and MobileNetV2 (Sandler et al., 2018).
>
> *Background*: Dynamic network inference is one of the techniques used to reduce the execution latency. The goal of this project is to determine whether adaptive neural network design is a promising way for more efficient network inference given modern neural networks. Please implement the approach in this paper, and identify the challenges and limitations of such an approach considering its hardware implications.

## 6. Student feedback

The course received positive evaluations from students during both initial and subsequent offering (averaging 4.4-4.6 on a scale of 1-5). Student comments reflect their positive experience and provide some insight into what makes for a balanced coverage of topics and suitable learning process. In general, the topics coverage was welcome in its breadth: "The course topics were sufficiently diverse and they covered both hardware, software, and hw-sw co-design approaches really well." In particular, the homework assignments were found to be well designed to enable learning and material understanding: "The assignments were not ridiculously long; hence, I had enough time to think through about the questions, improve my implementations, understand the essence of the field, and do extra readings on my own at times." or "Great class, I really liked the homework assignments." Overall, the course was viewed positively for how students were engaged in the learning process: "The courses were designed well to allow for good technical discussion in the class. I don't think that I participated in "active learning" by that degree in any other class."

## 7. Conclusion

We have designed a graduate-level course that focuses on the hardware aspect of modern machine learning. Due to the interdisciplinary nature of the course, we put focus on deep learning, adopt paper presentation sessions to foster discussions, design homework to prepare students for course projects and getting to learn the state-of-the-art, and provide predefined projects to help our students succeed. Our approach was deemed to work well empirically as the feedback from students was positive.

## Acknowledgement

We thank Ermao Cai, Zhuo Chen, and Ruizhou Ding for their help in putting up the course materials.

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
