# OpenReview forum: "Putting the "Machine" Back in Machine Learning for Engineering Students"
_ecmlpkdd.org/ECMLPKDD/2021/Workshop/TeachML — TeachML 2021_

### Official Review · Reviewer_QzMg · 2021-07-09
**review of: Putting the "Machine" back in machine learning for engineering students**

**Rating:** 8
**Confidence:** 4

**Review:**

This is a very interesting and well written paper covering teaching hardware architectures for machine learning. It should definitely be included in the conference.

A few minor comments:

* The abstract could do with an extra sentence covering the conclusion of the paper.
* The course seems to have a lot of content. What percentage of students total time does it take?
* In the hardware modeling section you state "We have provided a Python environment for the students which includes boilerplate code to interact with the hardware models.", what exactly is this python environment providing? Is it a hardware emulator? Which emulator?
* There is mention of FPGAs in table 1. Do students get to implement a machine learning algorithm on an FPGA? Do they have any prior FPGA experience before this?
* Were there any negative comments from students or aspects of the course that they struggled with?

---

### Official Review · Reviewer_PYV6 · 2021-07-16

**Rating:** 7
**Confidence:** 4

**Review:**

## Summary
This paper presents a report on a graduate level course covering hardware aspects of machine learning. Students learn to assess the hardware demands of their chosen models by using appropriate tooling both on the HW-SW side and the ML side. The paper describes its syllabus, the homework assignments and concludes with the overall (positive) feedback of their students.

I think the course design is really interesting (I would probably take that course) and the homework structure is helpful as an inspiration for other instructors. The paper is written as an experience report which makes it not highly obvious what to derive from it. I would have appreciated more insights into how students' feedback was incorporated, given that the course has been offered multiple times. Also, there were no didactic experiments that showcase the pedagogical efficacy or open educational resources (OER) shared along with the paper. Certainly, it adds a less frequently explored direction to the Teach-ML workshop which validates its acceptance.

## Minor points
* "While introductory courses for machine learning is ubiquitous" -> are

---

### Decision · Program_Chairs · 2021-07-21

**Decision:**

Accept

**Comment:**

Congratulations! The reviewers agree that this paper should be accepted.

Camera-ready version is due August 18, 2021. As you prepare the camera ready version, please take the reviewers comments into consideration.

We look forward to your participation at the workshop on September 13, 2021. We invite you also to join us for the satellite event on September 08, 2021. Schedules for both the workshop and the satellite event will be forthcoming.